# Embryonic Deletion of TXNIP in GABAergic Neurons Enhanced Oxidative Stress in PV+ Interneurons in Primary Somatosensory Cortex of Aging Mice: Relevance to Schizophrenia

**DOI:** 10.3390/brainsci12101395

**Published:** 2022-10-15

**Authors:** Ting Xue, Xiaodan Wang, Ying Hu, Ying Cheng, Han Li, Yuan Shi, Lijun Wang, Dongmin Yin, Donghong Cui

**Affiliations:** 1Shanghai Mental Health Center, Shanghai Jiao Tong University School of Medicine, Shanghai 201108, China; 2Shanghai Key Laboratory of Psychotic Disorders, Shanghai 201108, China; 3Shenzhi Department of the Fourth Affiliated Hospital of Xinjiang Medical University, Urumqi 830000, China; 4Key Laboratory of Brain Functional Genomics, Ministry of Education and Shanghai, School of Life Science, East China Normal University, Shanghai 200062, China; 5Brain Science and Technology Research Center, Shanghai Jiao Tong University, Shanghai 200240, China

**Keywords:** thioredoxin-interacting protein (TXNIP), developing brain, GABAergic, aging, schizophrenia, oxidative stress

## Abstract

The brain is susceptible to perturbations of redox balance, affecting neurogenesis and increasing the risks of psychiatric disorders. Thioredoxin-interacting protein (TXNIP) is an endogenous inhibitor of the thioredoxin antioxidant system. Its deletion or inhibition suggests protection for a brain with ischemic stroke or Alzheimer’s disease. Combined with conditional knockout mice and schizophrenia samples, we aimed to investigate the function of TXNIP in healthy brain and psychiatric disorders, which are under-studied. We found TXNIP was remarkedly expressed in the prefrontal cortex (PFC) during healthy mice’s prenatal and early postnatal periods, whereas it rapidly decreased throughout adulthood. During early life, TXNIP was primarily distributed in inhibitory and excitatory neurons. Contrary to the protective effect, the embryonic deletion of TXNIP in GABAergic (gamma-aminobutyric acid-ergic) neurons enhanced oxidative stress in PV^+^ interneurons of aging mice. The deleterious impact was brain region-specific. We also investigated the relationship between TXNIP and schizophrenia. TXNIP was significantly increased in the PFC of schizophrenia-like mice after MK801 administration, followed by oxidative stress. First episode and drug naïve schizophrenia patients with a higher level of plasma TXNIP displayed severer psychiatric symptoms than patients with a low level. We indicated a bidirectional function of TXNIP in the brain, whose high expression in the early stage is protective for development but might be harmful in a later period, associated with mental disorders.

## 1. Introduction

The brain is highly susceptible to oxidative stress compared to other organs. Oxidative stress-induced profound damage has been found to be widely involved in many neurodegenerative and neuropsychiatric disorders [1,2,3]. The thioredoxin (TRX) antioxidant system is one of the two major thiol-reductase systems which protects living cells from oxidative damage. The system comprises TRX, TRX reductase (TRX-R), thioredoxin-interacting protein (TXNIP), and NADPH (nicotinamide adenine dinucleotide phosphate). TRX has an anti-oxidative activity by interacting with the oxidized substrates to reduce their cysteine groups. TXNIP is an endogenous intracellular inhibitor of antioxidation that could bind to the active catalytic site of TRX to inhibit its oxidoreductase activity [4]. Hence, the overexpression of TXNIP could increase cellular oxidative stress.

TXNIP is widely expressed in almost all normal tissues [5], however, with lower expression in the healthy brain than in other organs [6]. The knowledge of its roles in the developing brain (a susceptible stage for oxidative stress [7]) is still limited to date. TXNIP could be induced by a variety of cellular stress that results in enhanced oxidative stress, such as UV light, H_2_O_2_, heat shock [8], high glucose [9], and NMDAR (N-methyl-D-aspartic acid receptor) hypofunction [10]. Besides its ability to enhance oxidative stress, increasing evidence has identified its diverse function, including inhibiting glucose uptake [9,11], inducing cellular apoptosis [12], and NLRP3 (NLR Family Pyrin Domain Containing 3)-mediated inflammation [13]. Hence, TXNIP has been recognized as a central hub linking different pathological processes, especially oxidative stress and inflammation [13].

Although the linkage of oxidative stress and inflammation has been suggested as one of the pathogeneses for neurodegenerative and neuropsychiatric disorders, how TXNIP is involved in these pathological processes is under-studied. Studies have identified increased TXNIP expression in brains of postmortem AD (Alzheimer’s disease) patients [14] and transgenic AD mice [15,16] compared to their controls. Depressive-like rodents induced by chronic unpredictable stress also exhibited an enhanced expression of TXNIP in their hippocampus and frontal cortex [17,18]. Interestingly, genetic ablation or pharmacological inhibition of TXNIP showed protective roles in reducing brain infarction in a mouse model of thromboembolic stroke [19] and improving behavioral performance in vascular dementia-like [20] or depressive-like mice [21], which could contribute to the reduced oxidative and inflammatory levels in the brain.

Parvalbumin interneurons (PVIs) form networks of inhibitory GABAergic synapses [22], contributing to excitation/inhibition (E/I) balance. They are also necessary for gamma oscillation [23], which is associated with higher brain functions [24]. PVIs exhibit fast-spiking firing properties, which renders them a higher energy demand. Thus, as a corollary, PVIs are more susceptible to oxidative stress [25] than other neurons. Oxidative stress has been identified as a common pathology leading to PVIs impairment in schizophrenia patients [26]. Gclm (Glutamate-cysteine ligase modifier subunit) knockout mice (a redox dysregulation model) also showed PVIs deficits in their anterior cingulate cortex (ACC) [27,28].

Considering the critical roles of TXNIP in linking oxidative stress and inflammation, our limited knowledge about its involvement in the central nervous system (CNS) makes it necessary further to investigate its function in healthy brain and psychiatric disorders. Here, we traced the expression pattern and cellular location of TXNIP in the developing brains of healthy mice. Next, we explored its function in the healthy brain based on transgenic mice with TXNIP conditional deletion in GABAergic neurons during the embryonic stage. We tested the resulting alteration in PVIs in ACC of aging mice. ACC (part of prefrontal cortex (PFC)) is sensitive to early life stress [29,30] and is affected in schizophrenia patients [31]. Finally, we detected the relationship between TXNIP and schizophrenia based on a previously established schizophrenia-like mice model and clinical samples, consisting of the first episode and during naïve (FEDN) patients and healthy controls. Overall, this study could provide insights into developing new therapeutics for brain therapy.

## 2. Materials and Methods

Animals

All procedures were approved by the Institutional Animal Care and Use Committee at Shanghai Jiaotong University and followed the National Institutes of Health guidelines. C57BL/6 wild type (WT) and *Txnip*^flox/flox^*Viaat*^cre/−^ mice were used. Housing conditions include a 12 h light/dark cycle, a 50 ± 5% humidity, and a controlled temperature of 22–25 °C in the Individual Ventilated Caging System (IVC, TECNIPLAST S.P.A., Buguggiate, Italy). Food and water were provided ad libitum. For experiments with C57BL/6 WT mice, two dosages of MK801 (0.5 mg/mL (10 mg/kg)) (Sigma-Aldrich, St. Louis, MO, USA, CAS No. 77086-21-6) or saline were intraperitoneally injected at an interval of 8 h (9:00 a.m. and 5:00 p.m.) on P10 (postnatal ten days), P17 (postnatal 17 days), P25 (postnatal 25 days), or 8 w (postnatal eight weeks), as needed.

Generation of *Viaat-Txnip* conditional knockout (KO) mice

Conditional *Txnip* knockout mice in GABAergic neurons (*Txnip^flox/flox^Viaat^cre/−^*) were generated by breeding *Txnip^flox/flox^* [32] with *Viaat^cre/−^* [33] mice, resulting in heterozygous *Txnip^flox/−^Viaat^cre/−^* mice. These mice were further bred with Txnip^flox/flox^ mice to obtain *Txnip^flox/flox^Viaat^cre/−^* (KO) mice. Littermates of *Txnip^flox/flox^Viaat^−/−^* were used as controls. *Txnip^flox/flox^Viaat^cre/−^* were born with the expected Mendelian ratio, had no detectable developmental defects, and were fertile. *Txnip^flox/flox^* and *Viaat^cre/−^* mice were kindly provided by Dr Dongmin Yin (East China Normal University). Both mice were backcrossed with C57BL/6 mice for at least six generations to purify their genetic background. Aging *Txnip^flox/flox^Vgat^cre/^*^−^ and their controls were sacrificed by 18 months.

Behavioral Tests

Behavioral tests related to schizophrenia, including open field test, nesting building, and Barnes maze, were conducted on P60-P70 of CD57BL/6 mice. Before behavioral tests, mice were acclimatized in the testing room for one hour, and tests were conducted bewteen 9:00 a.m. and 4:00 p.m. Detailed information is provided in our previous study [34].

Western blot

C57BL/6 mice were sacrificed, and PFC was dissected. Proteins were separated with 10% SDS-PAGE gel and blotted with primary antibodies as follows: rabbit anti-TXNIP (1:500, proteintech, Wuhan, China, RRID:AB_2876873), rabbit anti-β-actin (1:10,000, proteintech, Wuhan, China, RRID: AB_2923704), rabbit anti-GAPDH (1:10,000, Sigma, St. Louis, MO, USA, RRID: AB_2910561). The secondary antibody included goat anti-rabbit-IgG (1:10,000, Sigma, RRID:AB_92410).

RNA isolation and RT-PCR

Total RNA from frozen dissected PFC of C57BL/6 mice was extracted using RNAiso Plus (TaKaRa, Kusatsu, Japan, Cat. No. 9109). cDNA was synthesized from total RNA using the PrimeScript RT reagent Kit (TaKaRa, Cat. No. RR307A) according to the manufacturer’s introduction. Diluted cDNA was subsequently used for real-time RT-PCR using LightCycler 480 SYBR Green I Master (Roche, Basel, Switzerland, Cat. No. 04707516001). RT-PCR was performed in LightCycler^®^ 480 Instrument II (Roche). Three technical replicates were included. The primers were used as follows: mouse *Txnip*: -F: 5′-CCTAGTGATTGGCAGCAGGT-3′; -R: 5′-AAGGAGGAGCTTCTGGGGTA-3′; *Actin*: -F: 5′-AGGTCGGTGTGAACGGATTTG-3′, -R: 5′-TGTAGACCATGTAGTTGAGGTCA-3′.

Immunohistochemistry

Anesthetized aging *Txnip^flox/flox^Viaat^cre/−^* (KO) mice and controls were transcardially perfused with PBS followed by 4% paraformaldehyde (PFA). Brains were fixed overnight with 4% PFA, and dehydrated in 30% sucrose at 4 °C. Brains were sectioned coronally in 25 μm slices using a freezing microtome (Leica, Wetzlar, Germany). Sections between bregma 0.97 mm to 1.97 mm were collected. Sections were washed with PBS, blocked with 10% normal goat serum in 0.1% Triton-X-100 (CAS No. 9002-93-1) for two hours at room temperature (RT), and incubated with primary antibodies of anti-rabbit parvalbumin (1:2000, Swant, Barron, WI, USA, PV27a, RRID: 10000344), and anti-mouse 8-OHdG (1:200, Santa Cruz Niotechnology, Dallas, TX, USA, sc-393871, RRID: AB_2892631) at 4 °C overnight. Secondary antibodies were used as follows: Alexa 647 donkey anti-mouse IgG (1:1000, Jackson lab, Shanghai, China, RRID: AB_2340863) and Alexa 488 donkey anti-rabbit (1:1000, Jackson lab, RRID: AB_2313584) for 2 h at 37 °C. Cells were counted every sixth section. All images were obtained using thunder image systems (Leica).

RNA scope

In situ RNA analysis was performed using RNAscope^®^ Multiplex Fluorescent Reagent Kit v2 Assay (Advanced Cell Diagnostics, Hayward, CA, USA, Cat. No. 323100) according to the manufacturer’s introductions. Briefly, paraffin-embedded brains from C57BL/6 mice were cut into 5 μm and mounted on the Superfrost^®^ plus slides (Thermo Fisher Scientific, Waltham, MA, USA, Cat. No. 4951PLUS4). After pretreatments of hydrogen peroxide, target retrieval, and protease, slices were hybridized with probes targeting *Txnip* (Cat. No. 550901-C2), *CamkII* (Calcium/calmodulin-dependent protein kinase II, Cat. No. 445231), *Gad67* (Glutamic acid decarboxylase 67, Cat. No. 400951), and *Cd11b* (Integrin alpha-M, Cat. No. 311491), using a HybEz hybridization system. After several amplifications, sections were stained with DAPI and mounted with Prolong Gold^®^ Antifade Mountant (Thermo Fisher Scientific, Cat. No. P10144). Images were acquired using FV1200 confocal microscopes (Olympus, Tokyo, Japan).

Microglia isolation

Brains of C57BL/6 mice at P10 were harvested on ice, and PFC from 5 mice were dissected and mixed. Mixed PFC tissues were minced into small pieces (<1 mm) and digested in buffer (16.5 U/mL Papain, 125 U/mL DNase I in DMEM/F-12 with 15 mM HEPES) at 37 °C for 30 min. Digested tissues were transferred to a 70 μm nylon mesh strainer, and the cell suspension was collected. The 30% Percoll^®^ Solution (Solarbio, Beijing, China, CAS:65455-52-9) was added to the cell pellets to remove the myelin layer. Cells were then collected by centrifugation, and microglia were further isolated by EasySep Mouse CD11b Positive Selection Kit (StemCell, Vancouver, Canada, Cat. No. 18970) as the manufacturer’s introduction. Briefly, cells were incubated with CD11b antibody for 5 min at RT. Selection magnetic beads were added to samples and incubated for 5 min at RT. Microglia cells were then isolated with positive selection. 

Immunofluorescence 

Isolated microglia were seeded in a 24-well culture plate precoated with 0.1% sterile-filtered poly-L-Lysine (sigma, CAS No. 25988-63-0) and cultured in DMEM/F12 with 15 mM HEPSE (Gibco, MA, USA, Cat. No. C11330500BT) supplemented with 10% FBS (Gibco) at 37 °C overnight. Cells were then washed with PBS and fixed with 4% PFA for 30 min at 4 °C. After washing with PBS, cells were incubated with 0.3% Triton X-100 for 30 min and blocked with 5% normal goat serum for one hour at RT. After PBS washing, cells were incubated with primary antibody of anti-rabbit Iba1 (1:100, abcam, Cambridge, UK, 178847, RRID: AB_2832244) and anti-mouse TXNIP (1:100, abcam, Cat. No. ab210826) overnight at 4 °C. Secondary antibody of Alexa 594 donkey anti-rabbit (1:800, Jackson lab, RRID: AB_2340621) and Alexa 488 goat anti-mouse (1:800, Jackson lab, RRID: AB_233884) were used. Images were acquired using FV1200 confocal microscopes (Olympus). 

Proteomics analysis

Proteins of PFC from C57BL/6 mice at P10 were extracted and used for proteomics. Detailed information could refer to in our previous paper [34]. Synapt G2-Si quadrupole time-of-flight mass spectrometry (Waters Corporation) in UDMS^E^ mode was used for detection coupled with a nanoscale reverse-phase UPLC system. MS raw data were processed in Waters Progenesis QI (QIP, version 3.0.2). Proteins identified with at least two unique peptides and a fold change of ≥|1.3| were defined as differentially expressed proteins. The mass spectrometry proteomics data have been deposited to the ProteomeXchange Consortium (http://proteomecentral.proteomexchange.org (accessed on 5 July 2022)) via the iProX partner repository [35] with the dataset identifier PXD035089.

Participants 

The protocol was approved by the Institutional Review Board of Shanghai Mental Health Center (SMHC). Chinese Han FEDN patients were recruited from SMHC, and Chinese Han Healthy controls were recruited through the community from 2016 to 2020. The inclusion criteria of patients were: (1) met diagnosis of schizophrenia based on DSM-IV-TR (Text Revision); (2) aged from 18 to 75 years. Unrelated healthy controls aged 18 to 75 years and did not have a family history of mental disorders were included. Participants were excluded if they: (a) had any other major Axis I disorder; (b) had organic brain disorders and severe physical diseases; (c) were pregnant women. Finally, 126 FEDN patients and 478 healthy controls were recruited. All participants recruited provided their written informed consent. The psychiatric symptoms were evaluated using the Positive and Negative Syndrome Scale (PANSS). The plasma level of TXNIP was detected using an ELISA kit (Cat. No. SEB162Hu, Cloud-Clone, Corp., Houston, TX, USA). Briefly, 100 μL of diluted plasma was added to the wells and incubated for one hour at 37 °C. Afterward, 100 μL of detection reagent A was added and incubated for one hour at 37 °C. Wells were washed three times and incubated with detection reagent B for 30 min at 37 °C. After five washes of wells, 90 μL of substrate solution was added and incubated for 15 min at 37 °C in the dark. Then, 50 μL of stop solution was added and the measurement was conducted at 450 nm immediately. 

## 3. Results

### 3.1. TXNIP Was Highly Expressed in the Early Prenatal Period, Followed by a Rapid Decrease in PFC of Healthy Mice

Previous studies showed that the expression of TXNIP in a healthy brain is prominently lower than in other organs [6]. This phenomenon raised the question of the biological function of TXNIP in the healthy brain, which is under-studied. Here, we investigated the expression profile of TXNIP in PFC of healthy mice from E14.5 (Embryonic 14.5 days) to P405 (postnatal 405 days), representing the stage from embryonic to middle-aged. Interestingly, we found TXNIP was highly expressed during the prenatal period; however, it rapidly decreased from the first week (Figure 1a,b). Its expression during adulthood was lower compared to the prenatal period. We also investigated its mRNA expression in human postmortem brains through the BrainSpan atlas (http://www.brainspan.org/ (accessed on 20 September 2020)). Similarly, TXNIP was highly expressed in the early prenatal stage, sharply decreased after that, and maintained a quite low level throughout adulthood (Figure 1c). This expression pattern was coincident across multiple brain regions. These results drive our hypothesis that TXNIP may play an essential role in brain function during the early stage. 

### 3.2. The Induction of TXNIP Expression via NMDAR Hypofunction Was Only Detected at the Early Stage of Brain Development

To examine our hypothesis above, we performed a simple paradigm in which a median dosage of MK801 (0.5 mg/kg) was systematically administrated to mice on different postnatal days. MK801 was pharmacologically used as the inhibitor of NMDAR activity, whose dysfunction has been linked to cortical oxidative stress [10,36] and is relevant for the pathophysiology of many psychiatric disorders [37,38]. Previous studies have demonstrated that the inhibition of NMDAR activity via MK801 could deactivate the phosphatidylinositol-3 kinase-Akt (PI3K-Akt) pathway, trigging FOXO dephosphorylation. Dephosphorylated FOXO translocates to nuclear and increases the transcription of TXNIP [39,40] (Figure 2a). We found that increased TXNIP expression was observed at P10 (Figure 2b–e) but decreased from P17 after MK801 administration (Figure 2f–i). There was no significant induction of TXNIP expression at both P25 (adolescent stage) (Figure 2j–m) and 8 w (adulthood) (Figure 2n–q) for either mRNA or protein level. This result indicates that compared to the later developmental stage, the expression of TXNIP is more sensitive to cellular stress (e.g., NMDAR hypofunction) during the early postnatal period, suggesting a critical role of TXNIP in regulating the sensitivity window of anti-oxidative stress during brain development.

### 3.3. The Cellular Location of TXNIP in PFC during the Early Developmental Period

Previous studies [6,41,42] and recent single-cell sequencing of mice have identified the location of TXNIP primarily distributed in microglia, neuron, and astrocytes. However, these results are mainly focused on adult rodents. Since we aimed to explore the role of early expression of TXNIP in brain function, herein, we investigated its expression in PFC during different early postnatal days (from P4 to P10). We performed in situ RNA analysis using RNAscope, and we found Txnip mRNA sparkly distributed in GAD67 positive cells (GABAergic neurons) in PFC of P4 and P7 mice, especially at P4 (Figure 3a,b). However, its expression was not observed in GABAergic neurons at P10 (Figure 3c). In contrast, Txnip mRNA was expressed in CAMKII positive excitatory neurons across P4 to P10 (Figure 3d–f). We detected the scattered expression of Txnip mRNA in microglia at P7 using RNA scope (failed to detect at P10 and P17) (Figure 3g). After isolating microglia from PFC of P10 mice, we also identified protein expression of TXNIP via immunofluorescence staining (Figure 3h).

### 3.4. Embryonic Conditional Knockout of TXNIP Induced Increased Oxidative Stress in PVIs in S1 Region of Aging Mice

Although the expression of TXNIP in the early postnatal period was not cell-type-specific, this study mainly focused on its function in fast-spiking PVIs. PVIs are the primary class of GABAergic neurons and are highly sensitive to oxidative stress due to their higher demand for energy than other neurons [25]. Aging contributes to many physiological modifications, including declining higher cognitive processes such as memory functions. Higher cognitive functions depend critically on gamma oscillations (30–100 Hz), which results from the activity of PVIs. Altered gamma oscillations have been described in aged wild-type mice [43]. However, little is known about how embryonic deletion of TXNIP in PVIs affects its physiological state in aging mice. In this part, we applied the loxp-Cre system for conditional knockout of the expression of TXNIP during the prenatal stage in GABAergic neurons. We found no significant difference in the total number of PVIs, oxidative PVIs, and oxidative non-PVIs in the ACC region (data not shown) between aging TXNIP KO and control mice. However, we found that oxidative stress was mainly displayed in the S1 region (primary somatosensory cortex) in KO mice. Contrary to our expectation that TXNIP deletion might enhance the anti-oxidative ability in aging mice, in this region, the total number of cells with oxidative stress was significantly increased in KO mice compared to controls (Figure 4a–c). In addition, the number of oxidative PVIs and non-PVIs were also remarkably increased in KO mice (Figure 4d,i–k). The total number of PVIs and PVIs without oxidative stress were also significantly increased in KO mice (Figure 4e–h). These results indicate that, although TXNIP was an inhibitor of antioxidative stress, the typical thought of the protective effect of TXNIP deletion might be invalid during the embryonic stage, which in verse would advance the oxidative response in aging mice. Notably, this effect might be brain region-specific.

### 3.5. Increased Expression of TXNIP and Oxidative Stress in PFC of Schizophrenia-like Mice

Next, we investigated the expression of TXNIP in a schizophrenia-like mouse model. This model was established and published in our previous study, in which two transient intraperitoneal injections of MK801 (0.5 mg/kg) on P10 could induce schizophrenia-like behaviors in adolescent mice (~P35) [34]. This study expanded our behavioral test to adult mice based on the same paradigm (Figure 5a). MK801 injection did not affect mice weight (Figure 5b), travelling distance for both the inner and outer zones of the cage (Figure 5c,d), indicating that locomotion activity was not disturbed. We observed social deficits (the poor ability of nest building) (Figure 5e,f) and impaired learning and memory (longer latency in entering the escape chamber and less time exploring the target hole) in the adulthood of mice (Figure 5g,h), indicating schizophrenia-like behavior. In this paradigm, we found TXNIP was remarkably induced at 4 h post-MK801 injection; however, this induction rapidly recovered to the level comparable to the control groups (Figure 5i,j). Interestingly, proteomics analysis of PFC showed that the extreme induction of TXNIP did not immediately affect the expression of a set of proteins involved in anti-oxidative stress (Figure 5k–n). In contrast, this effect was displayed two weeks later (P25), in which proteins of LGUL (Lactoylglutathione lyase), GSTP1(Glutathione S-transferase P1), PARK7(Parkinson disease protein 7 homolog), and PYGL(Glycogen phosphorylase, liver form) were significantly decreased (Figure 5o–r). Their reduction indicates an increased level of oxidative stress in PFC. These results suggest that the increased expression of TXNIP associated with schizophrenia might induce enhanced oxidative pressure; however, it was time-dependent and may occur at a latency period.

### 3.6. The Plasma Level of TXNIP in FEDN Schizophrenia Patients

The schizophrenia-like mouse model provided preliminary knowledge about the association between TXNIP expression and schizophrenia. We involved 126 FEDN schizophrenia patients and 478 healthy controls in this part to investigate their relationships further. Due to the difficult accessibility of postmortem tissues, we applied plasma samples here. There was a significant difference in age, sex, and BMI at baseline between patients and healthy controls (Appendix A). After adjusting these covariates, we observed a significantly lower plasma TXNIP in FEDN patients compared to healthy controls (Figure 6a). However, when we further analyzed this result, we found that this significant discrimination could mainly result from a part of patients whose plasma level of TXNIP was much lower than that of any involved healthy individuals. We defined the threshold of plasma TXNIP of these patients as 1.76 ng/mL (Figure 6a). In this case, we further divided FEDN patients into two subgroups, i.e., FEDN (1), whose plasma level was above 1.76 ng/mL, and FEDN (2), whose plasma level was below 1.76 ng/mL. We found a significant difference in age and sex between FEDN (1) patients and healthy controls (Appendix A). After adjusting these covariates, there was no significant difference in plasma TXNIP levels between them (Figure 6b). In contrast, after adjusting the covariates of sex (Appendix A), FEDN (2) group still showed considerably lower expression of plasma TXNIP compared to healthy controls (Figure 6b). FEDN patients with a higher level of TXNIP showed significantly severer psychiatric symptoms (Figure 6d) than patients of the lower level, especially the positive symptoms (Figure 6c) after adjusting the covariates of BMI (Appendix A). However, the negative symptoms and general psychopathology were not affected (Figure 6e,f). These results suggest that for FEDN patients, higher plasma TXNIP levels could be associated with severer psychiatric symptoms.

## 4. Discussion

This study investigated the expression pattern of TXNIP in PFC of mice. It explored the biological effect of TXNIP on PVIs of aging mice through its conditional knockout in GABAergic neurons during the prenatal stage. Both are based on the healthy brain, which has not been thoroughly studied. Furthermore, we studied the relationship between TXNIP and schizophrenia using a schizophrenia-like mice model and clinical samples. By involving healthy and diseased brain samples, and animal and clinical cohort, our study provides further insight into the function of TXNIP in CNS and their roles in psychiatric disorders, which could facilitate its involvement in drug development for brain therapy.

This study demonstrated the expression pattern of TXNIP in PFC of healthy mice. We found TXNIP was only highly expressed in PFC during the prenatal and the first postnatal week, whereas this was lower during adulthood. Txnip mRNA expression from BrainSpan of human postmortem brain tissues supports our findings. What is the function of this expression pattern in the developing brain? TXNIP can inhibit the uptake of glucose. At birth, plasma glucose concentrations are 50% of the adult levels and achieve adult levels at P10 in rats. During the sucking period, the brain metabolizes glucose and ketone bodies [7]. Hence, we hypothesized that the high expression of TXNIP might help reduce the uptake and consumption of glucose in the fetal brain, resulting in reduced accumulation of ROS. Since infant animals have decreased levels of antioxidant enzymes/scavengers, such as glutathione peroxidase and glutathione [44,45], the reduced production of reactive oxygen species (ROS) might to protect infants’ brains from excessive oxidative stress. In contrast, adult brains achieve higher energy demand, and the decreased level of TXNIP could benefit their glucose uptake. Meanwhile, increased antioxidant enzymes could enhance their ability to resist increased ROS accumulation. On the other hand, our findings also demonstrate the possible reason why an infant period is a susceptible window for later psychiatric disorders than adulthood, emphasizing the importance of protecting the infant healthy. However, the detailed mechanism needs further investigation for us in the future.

This study demonstrated the cellular location of *Txnip* in mice PFC during the developing period using RNA scope. Previous studies through immunostaining or single cell-sequencing mainly focused on rodent adult brains, resulting in the most staining in microglia (https://portal.brain-map.org/ (accessed on 2 February 2021)) and some expression in neurons and astrocytes [6]. However, compared to microglia, we found more expression of *Txnip* in inhibitory and excitatory neurons in early postnatal days. This finding is similar to the human fetal brains from single-cell sequencing (http://www.psymukb.net/ (accessed on 20 February 2021) [46], http://stab.comp-sysbio.org (accessed on 10 May 2021) [47]). These results indicate that cell-type-specific expression of TXNIP is time-dependent, which could contribute to different brain function that is temporal and spatial dependent.

Studies focusing on the biological effect of TXNIP in the healthy brain are limited. ROS accumulation and increased oxidative stress have been demonstrated as one of the features of the normal aging brain. Previous studies showed that genetic ablation or pharmacological inhibition of TXNIP could reduce oxidative stress in a diseased brain, providing a protective effect for disease mitigation. However, contrary to our expectation, we found that conditional knockout of TXNIP in GABAergic neurons during the embryonic period was not protective but increased oxidative stress in both PVIs and other non-PVIs cells in normal aging mice brains. As hypothesized above, high prenatal expression of TXNIP might render the infant brain some defense to oxidative stress, while its reduction might destroy this balance. As studied, PVIs are highly vulnerable to oxidative stress early in development, resulting in increased permanent impairment in adulthood [27]. Interestingly, the observed impairment was brain region dependent, in which S1 was mainly affected compared to ACC. The primary somatosensory cortex, a part of the neocortex, receives the majority of general sensory signals for interpretation. Notably, along with the increased number of PVIs suffered from oxidative stress in KO aging mice, their number of non-oxidative PVIs were also increased. These phenomena might be a compensatory mechanism to maintain the E/I balance in the S1 region. Notably, Tauheed Ishrat et al. found that *Txnip^−/−^* aging mice showed a decreased level of NLPR3 mediated neuroinflammation compared to normal aging mice, which may protect the aging brain [48]. The difference in our observation might be due to different knockout mice models, targeted brain region, cell types, and pathways analyzed. Overall, our finding demonstrates that high expression of TXNIP in the early brain is essential for development, which is region dependent. The resulting redox dysregulation during the prenatal period could contribute to abnormal development, maturation, and impaired antioxidative ability in a later stage. However, our study could not demonstrate whether TXNIP deletion would result in abnormal electrophysiological properties of PVIs and behaviors in aging mice; this should be investigated in the future.

There are no studies about the relationship between TXNIP and schizophrenia. We provided a preliminary investigation using a previously established schizophrenia-like mouse model. Contrary to the depressive-like mice, whose TXNIP was kept increasing during their behavioral tests [17,18], we only observed a transiently rapid upregulation of TXNIP in PFC post-MK801 injection. This difference could attribute to a different degree of stress exposure. We hypothesized that a persistent injection of MK801 (such as 7 days or more) instead of two injections might generate schizophrenia-associated lasting upregulation of TXNIP; however, this should be investigated in the future. In addition, this transient upregulation was also observed in another study of cerebral venous sinus thrombosis in rats [49], suggesting the time for increased expression will be limited to ongoing active sites of pathology. The limitation of this part is that we could not demonstrate the causal effect between TXNIP and schizophrenia. Blocking the expression of TXNIP is needed to investigate whether schizophrenia-like behaviors could be recovered. However, this is the aim of this preliminary study.

Although lacking postmortem tissues, our preliminary study examined the relationship between the plasma level of TXNIP and schizophrenia. We found that approximately 75% of FEDN schizophrenia patients displayed a relatively high plasma level of TXNIP. Interestingly, these patients represented significantly severer psychiatric symptoms compared to patients with a low plasma level of TXNIP. We hypothesized that these highly expressed patients might have an increased number of damaged neurons, which could secrete more TXNIP to the circulation system, resulting in poorer psychiatric symptoms. Hence, our results indicate that antioxidant medicine could serve as adjuvant therapy for these patients. The limitation of the part is the number of FEDN patients was much fewer than heathy controls, which could result in some interpretation bias. However, recruiting sufficient FEDN patients is a common challenge in the clinic. On the other hand, our sufficient number of healthy controls could provide a precise threshold of TXNIP level to discriminate FEDN subgroups. In the future, we need to follow up on the expression of TXNIP in patients during the prodromal phase, which could help us better understand its application as a biomarker of schizophrenia.

## 5. Conclusions

Overall, we first demonstrated the bidirectional function of TXNIP in the brain, whose high expression during the early stage could be protective for brain development. Its sustained high level after birth could result in harmful impacts associated with mental disorders such as schizophrenia.

## Figures and Tables

**Figure 1 brainsci-12-01395-f001:**
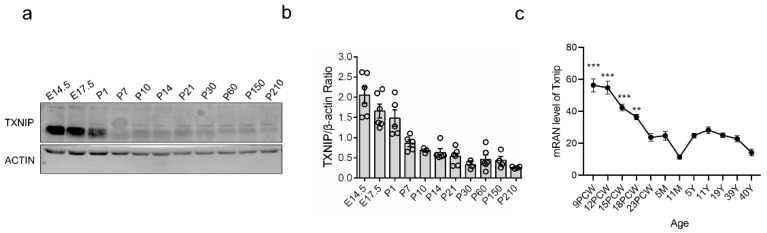
Expression pattern of TXNIP (Thioredoxin-interacting protein) underlying normal and NMDAR (N-methyl-D-aspartic acid receptor) hypofunction stage. (**a**) Western blot illustration of TXNIP expression pattern in mouse PFC (prefrontal cortex) from E14.5 (embryonic 14.5 days) to P405 (postnatal 405 days). (**b**) Statistic results of protein expression ratio between TXNIP and β-actin (internal standard) in mouse PFC. n = 3~6 for E14.5 to P210. (**c**) Gene expression pattern of *Txnip* in different regions of the human brain. Data were extracted from Brainspan (http://www.brainspan.org (accessed on 20 September 2020)) and displayed in the figure by the author. PCW: post-conceptual weeks. M: month. Each point represents one brain region, including ITC: inferior parietal cortex; A1C: primary auditory cortex; AMY: amygdaloid complex; CBC: cerebella cortex; DFC: dorsolateral prefrontal cortex; HIP: hippocampus; IPC: posteroventral (inferior) parietal cortex; M1C: primary motor sensory cortex; OFC: orbital frontal cortex; S1C: primary somatosensory cortex; STC: posterior (caudal) superior temporal cortex; STR: striatum. Statistics were performed between 9PCW vs. 5M, 12PCW vs. 5M, 15PCW vs. 5M, and 18PCW vs. 5M. Data were represented as mean ± SEM. *** *p* < 0.001; ** *p* < 0.01.

**Figure 2 brainsci-12-01395-f002:**
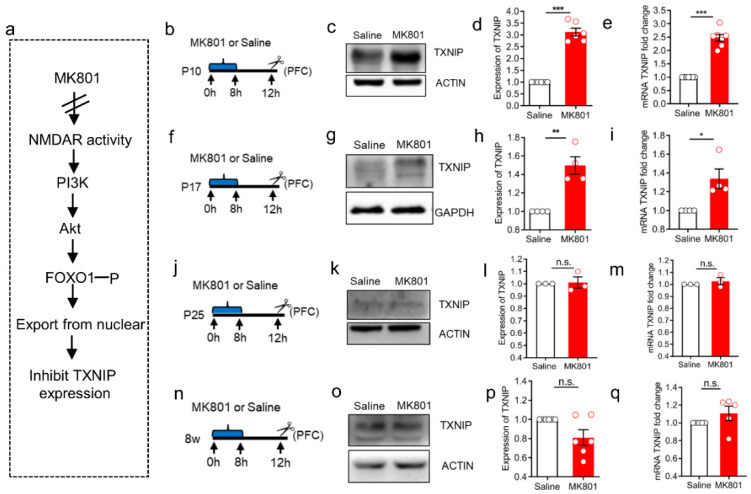
Expression pattern of TXNIP underlying NMDAR hypofunction. (**a**) Schematic illustration of TXNIP expression regulated by NMDAR activity. Inhibition of NMDAR activity via MK801 could deactivate the phosphatidylinositol-3 kinase-Akt (PI3K-Akt) pathway, trigging FOXO dephosphorylation. Dephosphorylated FOXO translocates to nuclear, binds to Txnip promoter, and increases its transcription. (**b**,**f**,**j**,**n**) Schematic graphs of MK801/Saline administration on P10, P17, P25, and 8 w. Two dosages of MK801 or Saline were injected at an interval of 8 h. PFC of C57BL/6 mouse was collected 4 h after the last injection. n = 3–8. (**c**,**g**,**k**,**o**) Western blot analysis of TXNIP expression in PFC 4 h after the last injection of MK801 or Saline on P10, P17, P25, and 8 w. (**d**,**h**,**l**,**p**) Statistic results of the protein expression ratio of TXNIP compared between MK801 and Saline-treated groups. (**e**,**i**,**m**,**q**) RT-PCR results of *Txnip* mRNA level compared between MK801- and Saline-treated groups 4 h after the last injection on P10, P17, P25, and 8 w. Data were represented as mean ± SEM. *** *p* < 0.001; ** *p* < 0.01; * *p* < 0.05; n.s., no significance.

**Figure 3 brainsci-12-01395-f003:**
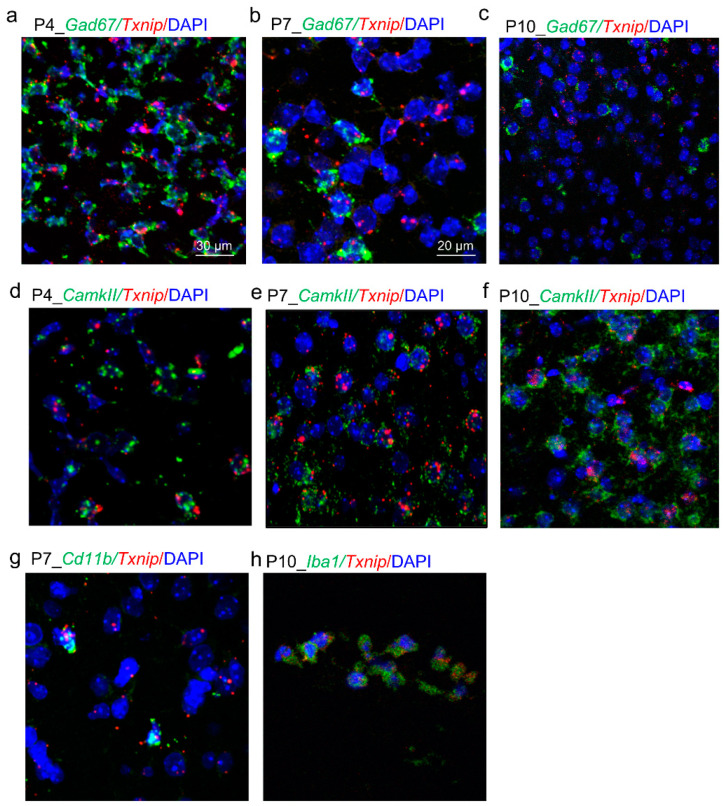
Cellular location of *Txnip* mRNA during development in mouse mPFC (medial prefrontal cortex). (**a**–**c**) Representative images of *Txnip* mRNA expression in GAD67 (Glutamic acid decarboxylase 67) positive neurons from P4 to P10 in mPFC using RNA scope. (**d**–**f**) Representative images of *Txnip* mRNA expression in CAMKII (Calcium/calmodulin-dependent protein kinase II) positive neurons from P4 to P10 in mPFC using RNA scope. (**g**) Representative images of *Txnip* mRNA expression in CD11b (Integrin alpha-M) positive cells in mPFC of P17 mouse using RNA scope. (**h**) Representative image of TXNIP protein expression in CD11b positive cells in cultured neurons isolated from PFC of P10 mouse using immunofluorescence staining. Scale bar, 30 μm (**a**), 20 μm (**b**–**h**).

**Figure 4 brainsci-12-01395-f004:**
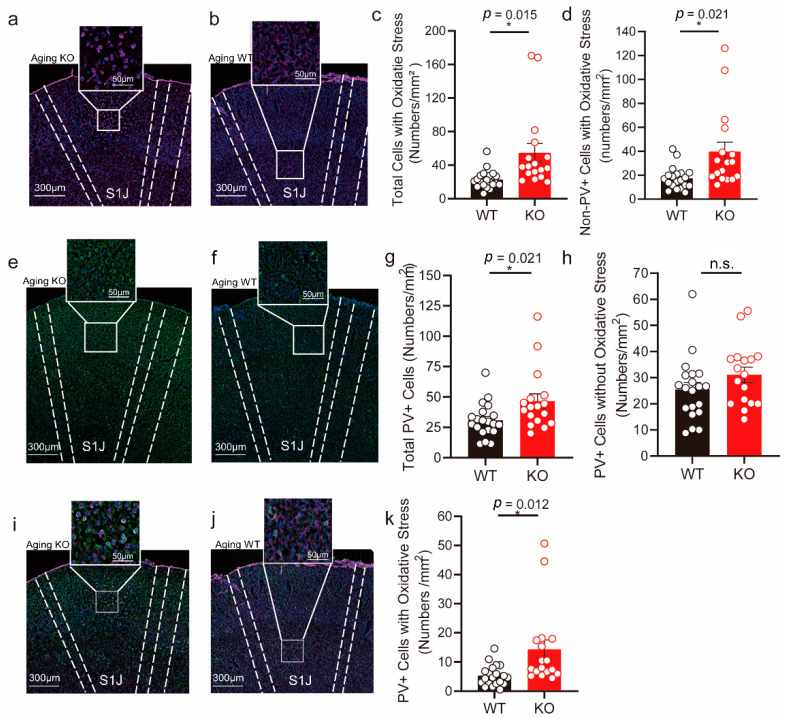
Aging mice with TXNIP conditional knockout (KO) in PVIs (Parvalbumin interneurons) during the prenatal period showed increased oxidative stress in the S1 region. (**a**,**b**) Representative images of oxidative stress marker of 8-OHdG staining (purple) in the S1 region. **Left**: TXNIP KO groups. **Right**: WT (wild type) groups (littermates). (**c**) Cell density with 8-OHdG positive staining in the S1 region. (**d**) The density of non-PVIs with 8-OHdG positive staining. (**e**,**f**) Representative images of PVIs (green) in the S1 region. **Left**: TXNIP KO groups. **Right**: WT groups (littermates). (**g**) The density of PVIs in the S1 region. (**h**) The density of PVIs cells without 8-OhdG staining. (**i**,**j**) Representative images of PVIs (green) with oxidative stress (purple) in the S1 region. **Left**: TXNIP KO groups. **Right**: WT groups (littermates). (**k**) The density of PVIs with oxidative stress. n = 4 mice for each group. N = 3*–*5 slices for each mouse in KO group, N = 4*–*5 slices for each mouse in the WT group. Cell numbers were calculated by the average of two hemispheres of the S1 region. Scale bar: 300 μm, 50 μm (zoom in the region). Data were represented as mean ± SEM. * *p* < 0.05; n.s. no significance.

**Figure 5 brainsci-12-01395-f005:**
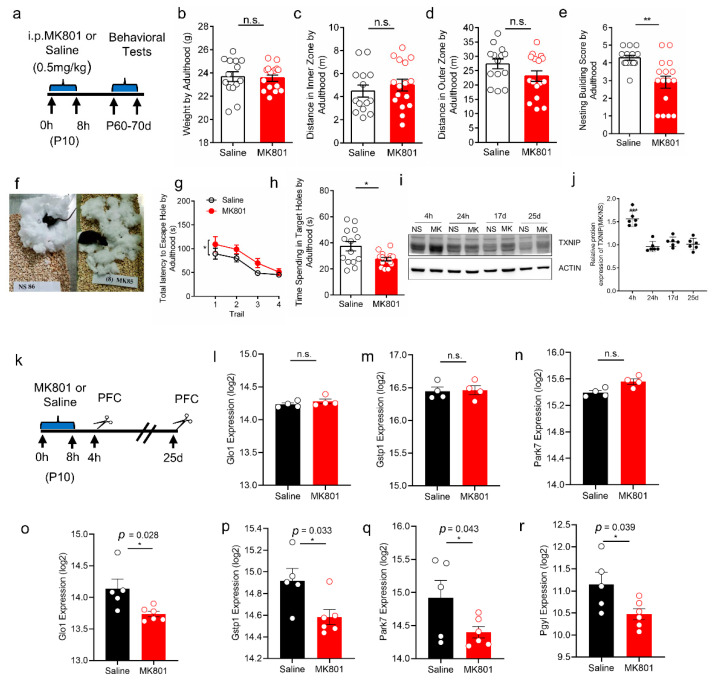
TXNIP expression and oxidative stress displayed in schizophrenia-like mice models. (**a**) Schematic graph of schizophrenia-like mice model. P10 mice were intraperitoneally injected with two dosages of MK801 (0.5 mg/kg) or saline with an interval of 8 h. Behaviors were tested during P60-70 d (adult). (**b**) Mice weight on P60 prior to the behavioral test, *p* = 0.8002. (**c**) Travelling distance in the center zone of the cage, *p* = 0.4709. (**d**) Travelling distance in the outer zone, *p* = 0.1074. (**e**) Nesting building score was significantly lower in MK801 treated mice, *p* = 0.0014. (**f**) The nesting building image between MK801 treated and control mice. (**g**) MK801 treated mice showed longer latency in entering the escape chamber (F (1, 108) = 5.416, *p* = 0.0218, Two-way ANOVA). (**h**) MK801 injected mice spent significantly less time exploring the target hole, *p* = 0.01. Data were represented as mean ± SEM. n = 14 for saline and n = 16 for MK801. * *p* < 0.05, ** *p* < 0.01. Student’s *t*-test for two groups’ comparison. (**i**) TXNIP expression post-MK801 injection by 4 h, 24 h, 17 d, and 25 d. NS: saline group. MK: MK801 group. (**j**) Statistic results of TXNIP expression ratio relative to saline group by Western blot. n = 6 for each group and time point. (**k**) Schematic graph of MK801 or saline administration and PFC collection. Saline or 0.5 mg/kg MK801 were intraperitoneally injected with two dosages of MK801 (0.5 mg/kg) or saline with an interval of 8 h as the schizophrenia-like model. PFC was collected 4 h and 25 d after injection. Proteins were extracted from PFC and subjected to proteomics analysis. n = 4. (**l**–**n**) Protein expression (log2 intensity) involved in anti-oxidative stress detected by mass-spectrometry 4 h after injection. (**o**–**r**) Protein expression (log2 intensity) involved in anti-oxidative stress detected by mass-spectrometry 25 d after injection. Data were represented as mean ± SEM. ** *p* < 0.01; * *p* < 0.05; n.s. no significance.

**Figure 6 brainsci-12-01395-f006:**
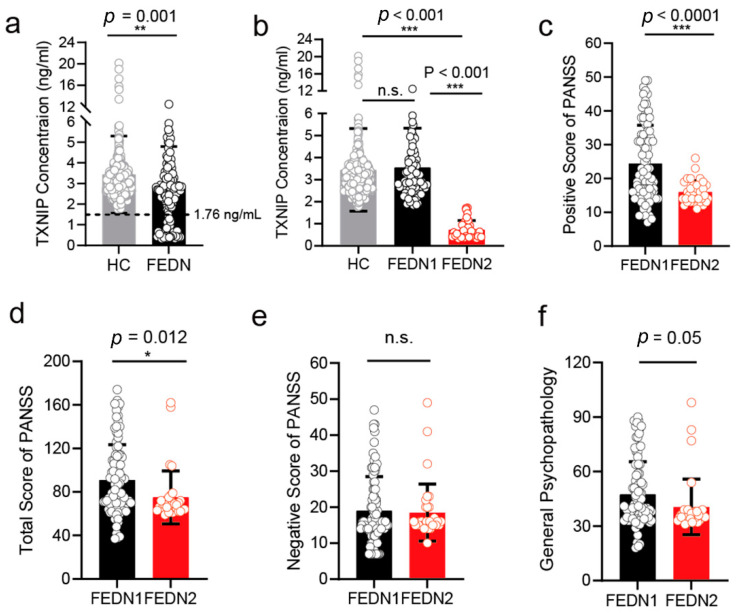
Plasma TXNIP expression in the first episode and drug naïve schizophrenia patients and healthy subjects. (**a**) The plasma TXNIP level compared with the first episode and the drug naïve (FEDN) schizophrenia patients and healthy subjects. The 1.76 ng/mL threshold was defined as the threshold to discriminate between high and low levels of TXNIP in FEDN schizophrenia patients. (**b**) Plasma TXNIP level was compared between high level, low level, and all healthy subjects. (**c**–**f**) subscale and total score of PANSS (Positive and Negative Syndrome Scale) compared between the low and high levels of plasma TXNIP groups. Data were represented as mean ± SEM. *** *p* < 0.001; ** *p* < 0.01; * *p* < 0.05; n.s. no significance.

## Data Availability

The data presented in this study are available on request from the corresponding author. The proteomics data have been deposited to the ProteomeXchange Consortium (http://proteomecentral.proteomexchange.org (accessed on 5 July 2022)) via the iProX partner repository [35] with the dataset identifier PXD035089.

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
