# Peer review of "Embryonic Deletion of TXNIP in GABAergic Neurons Enhanced Oxidative Stress in PV+ Interneurons in Primary Somatosensory Cortex of Aging Mice: Relevance to Schizophrenia"

_brainsci, 2022, doi:10.3390/brainsci12101395_

Round 1

Reviewer 1 Report

The manuscript titled with “Embryonic Deletion of TXNIP in GABAergic Neurons Enhanced Oxidative Stress in PV+ Interneurons in Primary Somatosensory Cortex of Aging Mice: Relevance to Schizophrenia” have shown that developmental expression of TXNIP in PFC. TNXIP deletion in PV neurons increased oxidative stress in schizophrenia-like mice. The manuscript is very well designed but poorly written and needs to increase the N in general. There are some issues need to address.

Major issues:

1.     The manuscript title is based on the Figure 4 data that embryonic deletion of TXNIP in GABEergic neurons increased oxidative stress in PV neurons in S1. The data in Fig.4 graphs have high variability. The observed significant changes might because of the outliers that are on the top. They are 3X to 4X higher values than others. How authors would justify it?

2.     N should be increased to minimum 6 animals in all figures.

3.     The blots in Fig.2C and 2G are on the same blots? If not, they are not comparable. So, it is ideal to say that increased TNXIP expression was observed at P17 but decreased from P10.

4.     What are the control experiments have done to confirm the specificity of GAD67 probes? Because at P4, almost all the cells are in green color. During the first postnatal week, GAD67 expressed only in few neurons and increased with time.

Minor issues:

Provide CAS number for all chemicals and reagents used in the study.

Line 26: expand GABA and use abbreviation throughout manuscript

Line 43: expand NADPH

Line 52: expand NMDAR

Line 59: Pathegeneses should be corrected to “Pathogenesis” 

Line 75: expand Gclm

Line 98: What is P10, P17, P25, 8W? I think P in P10 represents postnatal day. It should be defined and also same for 8W.

Line 152: probes should mention in italics and also check throughout manuscript

Line 160: HEPSE should be corrected to HEPES

Line 214-215: At P7, TXNIP is already down. It is ideal to say decreased expression from first week.

Line 248: age should be defined at their first appearance

Line 254-280: Fig.2 legend logically re-written to avoid the repetition of words. Fig.2b, 2f, 2j and 2n are not explained in the figure legends as shown in the figure.

Line 294: Scale bar is not necessary to show on every image If the images are having same resolution

Line 329: Label the colors on the image, what are the colors they represent? What is the age of Fig.4? Authors can increase the size of the zoomed images.

Line 547-656: The authors should carefully check the formatting of the references. There are many references with capital letters in the title, and others not.

Reviewer 2 Report

This is an extensive work showing evidence regarding the possible involvement of the thiorodoxin interacting protein (TXNIP) in Schizophrenia. The context and hypothesis are very well explained, and it is an easy to read paper despite the high amount of techniques applied.

However, there are a few suggestions for the authors to improve even more the quality of the present work:

1. Please beware of the lack of explanation of several abbreviations (such as FEDN, first-episode frug naive schizophrenia patients, which I only found described in the supplementary data ). Please review that all the abbrevations are explained at some point of the paper.

2. Figure1C is a bit confusing, since they separate the brain regions by colors that do not let the reader see clearly and they do not apply any statistics. On the other hand, they do not carry out the RT-PCR by themselves, but take it from the BrainSpan atlas. I strongly recommend to the authors to represent the Figure in a clearer way and apply statistics. If they have access to human postmortem brain, it would be good to investigate the reproducibility of the results obtained in the brain atlas.

3.  Page 4/17, Immunofluorescence part, line 8: please correct the typo: AlexA 594. Page 7/17, section 3.3: please correct the typo "discributed in microglia, neuron anD astrocytes". Page 12/17, last sentence: please remove "did" (the sentence "did was not protective" is not gramatically correct). Page 13/17 line 475: add a "be" in "our observation might BE due to...:". Line 485: mouse modeL. Line 507: sentence: "antioxidant medicine could serve". Please remove the "s" of "serves". Line 507: is a common challenge (instead of "challenging"). Please check other possible typos/grammar issues.

4. Figure 4: it is strongly recommended to improve the quality of the images showing the PVI cells by increasing the size and/or the magnification. Otherwise, it is hard to see, even though the quantification is shown.

5. Please explain in the methods how the TXNIP plasma levels are measured in human samples, since no ELISA or any other method are explained.

6. Please explain why in the methods all the cell culture is related to microglia, whereas in the text the authors refer to neurons (for instance, in page 8/17, line 1). If I understood correctly, the authors explain how microglia is isolated and cultured - selected as CD11b positive cells - and then they talk about cultured neurons. Please clarify and/or correct where needed.

Reviewer 3 Report

Dear Editor,
I really appreciate the opportunity to review the manuscript brainsci-1928396 entitled:
"Embryonic Deletion of TXNIP in GABAergic Neurons Enhanced Oxidative Stress in PV+ Interneurons in Primary Somatosensory Cortex of Aging Mice: Relevance to Schizophrenia"

Excellent work; my only comment concerns the choice of patients to be included. The use of the DSM-IV, which is from 1994, almost 30 years ago, is reported. Given that it would have been better to use DSM-5, now from 2013, I guess the authors meant DSM-IV-TR (Text Revision), from 2000, Please, check and eventually correct.

Author Response

Response: We thank the reviewer’s correction. We used the DSM-IV-TR and revised it in Line 208-209. 

Round 2

Reviewer 1 Report

Authors answered all, except increasing the N number, the issues.